# The Closure Relations in High-Energy Gamma-ray Bursts Detected by Fermi-LAT

Maria Dainotti [1,2,3,*], Delina Levine [4], Nissim Fraija [5], Donald Warren [6], Peter Veres [7], Shashwat Sourav [8]

1    National Astronomical Observatory of Japan, 2-21-1 Osawa, Mitaka, Tokyo 181-8588, Japan
2    Space Science Institute, Boulder, CO 80301, USA
3    Department of Astronomical Science, The Graduate University for Advanced Studies, SOKENDAI, Shonankokusaimura, Hayama, Miura District, Kanagawa 240-0193, Japan
4    Department of Astronomy, University of Maryland, College Park, MD 20742, USA
5    Instituto de Astronomía, Universidad Nacional Autónoma de México Circuito Exterior, C.U., A. Postal 70-264, Mexico 04510, Mexico
6    RIKEN Interdisciplinary Theoretical and Mathematical Sciences Program (iTHEMS), Wakō, Saitama 351-0198, Japan
7    Center for Space Plasma and Aeronomic Research, University of Alabama in Huntsville, 320 Sparkman Drive, Huntsville, AL 35899, USA
8    Department of Engineering Sciences, Indian Institute of Science Education and Research, Bhopal 462066, India
*    Correspondence: maria.dainotti@nao.ac.jp

**Abstract:** Gamma-ray bursts (GRBs) are brief, intense pulses of high-energy emission associated with extreme astrophysical phenomena, e.g. the death of massive stars or the coalescence of compact objects. They have been observed at high energies by the Fermi Large Area Telescope (LAT), which detects GRBs in the 20 MeV–300 GeV energy range. The Fermi-LAT Second GRB Catalog (2FLGC) presents information on 186 GRBs observed from 2008 to 2018. We consider the GRBs that have been fitted in the 2FLGC with a broken (21 GRBs) or simple power law (65 GRBs), compiling a total sample of 86 GRBs. We analyze the relationship between the spectral and temporal indices using closure relations according to the synchrotron forward-shock model evolving in stratified environments ($n \propto r^{-k}$). We find that the model without energy injection is preferred over the one with energy injection. There is a clear preference for the cooling conditions $\nu > \max\{\nu_c, \nu_m\}$ and $\nu_m < \nu < \nu_c$ (where $\nu_c$ and $\nu_m$ are the cooling and characteristic frequencies, namely the frequency at the spectral break). Within these cooling conditions, density profiles $r^{-k}$ with values of $k = 1.5$ and 2 generally have a higher rate of occurrence when considering relations with and without energy injection.

**Keywords:** gamma-ray burst; general–methods; data analysis

## 1. Introduction

Gamma-ray bursts (GRBs) are highly energetic events typically detected in the MeV-GeV energy range, although their energies can range as high as TeV. GRBs are classified according to the duration of their initial burst of emission, called the prompt emission. This emission is usually observed in the keV-MeV energy range, and their spectrum is generally described with a phenomenological function, such as the so-called Band functions [1], or with multiple functions including a thermal component [2,3]. The long-lasting emission after the prompt emission is called the afterglow phase, typically observed from radio to soft X-rays (see [4], for a review). However, in many cases, this extended emission can be detected in the MeV-GeV range, with one of the first examples of this behavior seen in GRB 090510 [5]. Traditionally, GRBs are classified according to their duration, $T_{90}$, the time in which the burst emits from 5% to 95% of its total energy in the prompt emission. Short GRBs (sGRBs) are classified as $T_{90} < 2$s, whereas long GRBs (lGRBs) have $T_{90} > 2$ s.

The best resource to crucially investigate the MeV-GeV emission is the Large Area Telescope (LAT) instrument aboard the Fermi Gamma-ray Space Telescope (hereafter Fermi-LAT, [6]), capable of measuring high-energy GRBs in the 20 MeV–300 GeV range. The First [7] and Second LAT GRB Catalogs [8] extensively analyze high-energy GRBs. Our study here refers to the Second Fermi-LAT GRB Catalog (2FLGC), based on GRBs detected by the Fermi-LAT instrument from 4 August 2008 to 4 August 2018. The catalog reports 186 bursts with emissions $\geq$ 30 MeV; 91 GRBs were detected in the 20–100 MeV energy range, and 169 bursts showed emissions in the range of 0.1–100 GeV. Among those, only 29 bursts were detected $\geq$ 10 GeV.

Several of those high-energy bursts— GRB 090510, GRB 090902B, GRB 090926A, GRB 130427A, GRB 131108A, GRB 160509A, and GRB 160821A—have peculiar traits, such as unusually high fluences in $\gamma$-rays [8]. GRB 090510 is a short GRB with a very high energy afterglow [5], whereas the others are long GRBs. GRB 131108A presented the first observational evidence of high-latitude emission at energies in the GeV range, also evolving unexpectedly, with a steep decline in emissions followed by three bright flares before a much longer-lasting decay [9]. GRB 160509A had extensive follow-up observations in multiple radio frequencies, which thus presented the second multi-wavelength observation of a reverse shock within a radio afterglow [10]. GRB 160821A is the third-most energetic burst observed with Fermi-LAT and provided the first evidence for high and varying polarization [11]. Another notable burst is GRB 130427A, which is the highest-energy burst (in both fluence and flux) detected by Fermi-LAT thus far [8]. For the cases of GRB 090510, GRB 090902B, and GRB 160609, Dainotti et al. [12] found that these GRBs show an indication of the plateau emission and obey the fundamental plane relation between the luminosity at the end of the plateau emission, its rest frame duration, and the peak prompt luminosity [13–17].

The most widely accepted model for the prompt emission and long-lasting afterglow phase is the standard fireball model [18–22]. The long-lasting emission is interpreted as originating when the relativistic outflow sweeps up the external medium [20,21]. During the external forward shock (FS), electrons are accelerated and cooled mainly by synchrotron radiation. One of the critical features of the standard fireball model is the generation of the FS, which can be tested with closure relations (CRs). Within the standard fireball model, these are equations that define the relationship between the temporal index, $\alpha$, and the spectral index, $\beta$, according to the convention $F_\nu \propto t^{-\alpha} \nu^{-\beta}$. These equations are derived using the assumptions of possible astrophysical environments, including the uniform density interstellar medium (ISM) or stellar wind environments, and the cooling regimes of these GRBs.

The constant-density ISM is assumed for sGRBs [23], whereas a stellar wind environment is assumed to be due to the collapse of massive stars [24–27]. Usually, studies of GRB CRs focus on the ISM and stellar wind environments with $k = 0$ and $k = 2$. However, numerous cases have been found of environments with $n(r) \propto r^{-k}$, with $k$ between 0 and 2.9 [28–33].

Prior studies of *Swift* X-ray light curves have shown more complex features than a simple power-law decay [34–44], including a plateau, thought to be produced by continuous energy injection. This plateau occurs immediately after the prompt emission and before the decay phase and has been found in X-ray [35–37,42,45], optical [17,46], and radio wavelengths [47].

Adopting a similar model of the afterglow for high-energy light curves, in our study, we test CRs for the high-energy ($\geq$100 MeV) temporally extended emission, which may last from hundreds to thousands of seconds compared to the short-lived emission of the GRBs observed at lower energies. It is usually described by the synchrotron FS model and hence is expected to satisfy its CRs [48,49]. Tak et al. [50] performed a systematic analysis of the CRs in a sample of 59 bursts with an error on their temporal index of less than 1/2 and an error on their spectral index of less than 1/3. They found that, although the standard synchrotron FS emission describes the spectral and temporal indices in most cases,

a considerable fraction of bursts cannot be described satisfactorily within this model. The LAT-detected bursts that do not satisfy any CRs exhibit a temporal decay index $\alpha_{\mathrm{LAT}} < 1$, corresponding to relatively shallow decay. They also found that several bursts fulfill the CRs of the slow-cooling regime [1] ($\nu_m < \nu_{\mathrm{LAT}} < \nu_c$), where $\nu_m$ and $\nu_c$ are the characteristic and cooling spectral breaks, respectively (see Sari et al. [23]). In addition, they found that lGRBs fulfilled the CRs derived in the constant-density medium.

However, one can argue that it is more likely for high-energy emissions above 100 MeV to be naturally produced by the synchrotron self-Compton (SSC) process, rather than synchrotron emission. Photon energies less than a few GeV are hard to explain with synchrotron emission. The synchrotron limit is from hundreds of MeV to a few GeV (for a full explanation, see Fraija et al. [51], Warren et al. [52], or Veres and Mészáros [53]). Therefore, it would not be surprising if many GRBs do not fit CRs based on synchrotron emission, as we show in our results.

In high-energy emissions, a previous study by Dainotti et al. [12] considered CRs for three GRBs (090510, 090902B, and 160509A). The authors found that all three GRBs satisfied an slow-cooling regime over a fast-cooling regime, either in a constant-density ISM or a wind medium.

Regarding the study in X-rays, Racusin et al. [54] considered CRs both with and without energy injection for the ISM and stellar wind environments to identify jet breaks within their light curve sample. Srinivasaragavan et al. [55] investigated the CRs of 455 *Swift*-observed light curves after the end of the plateau emission. They found good agreement with the external FS model in most cases, with the slow-cooling regime preferred in both the ISM and stellar wind environments. Using the same sample, Dainotti et al. [56] studied the compatibility of the X-ray light curves within the plateau region with the external FS model, accounting for energy injection with the standard fireball model, finding agreement in the majority of cases. They similarly found the most preferred environment to be the stellar wind in the slow-cooling regime, with the slow-cooling ISM environment as the second most preferred. Other studies have conducted a similar analysis of CRs between X-ray and optical data. Ackermann et al. [57] conducted a multiwavelength study of the LAT-observed GRB 110731A and determined that both the X-ray and optical afterglows favored a stellar wind environment in the slow-cooling regime.

Wang et al. [58] conducted a multi-wavelength study of 85 *Swift*-observed GRBs, analyzing the CRs of the X-ray and optical light curves to determine their compatibility with the external FS model. They found that 45 out of 85 GRBs presented an achromatic break and agreed with the standard CRs of the FS model in all segments of the afterglow. They found an additional 37 GRBs that disagree with the CRs in one or more afterglow segments, but present an achromatic break, suggesting that most of their sample can be described in part by the FS model. Fukushima et al. [59] conducted a numerical simulation of afterglow emission in X-ray and optical, considering the X-ray CRs with an ISM environment in both the fast-cooling and slow-cooling regimes. They applied their model to GRB 130427A, finding differences between the decay indices in X-ray and optical that suggest a more complicated model is needed to reflect the light curves' morphology accurately.

Further studies of CRs have included radio data as well. Kangas and Fruchter [60] conducted a multi-wavelength study of 21 *Swift* GRBs, comparing the behavior of the radio light curves to those in X-ray and optical. They found the radio light curves to be largely incompatible with the observed behavior of the X-ray and optical light curves and, therefore, inconsistent with the standard model. Misra et al. [61] studied the multi-wavelength afterglow of GRB 190114C and similarly found the radio and X-ray light curves incompatible with the standard model, suggesting a more complex analysis would be needed for an accurate understanding of the afterglow behavior.

In this paper, we expand on the findings of Tak et al. [50] with a sample of 86 GRBs observed by LAT for which the light curves can be fitted with a power-law (PL; 65) or with a broken power-law (BPL; 21) function presented in the 2FLGC [8]. We also examine the possibility of energy injection using a new set of CRs with different values of $k = 0, 1, 1.5, 2,$

and 2.5, chosen to evenly sample the possible space between $k = 0$ and $k = 2.9$. The structure of this paper is as follows: in Section 2 we explain the sample used for our analysis; in Section 3 we test the theoretical CRs for a general stratified circumburst medium and astrophysical environments used in this study along with our process for determining consistency with these relations, and detail our results; and in Section 4 we compare our results to prior results in other bands. Finally, in Section 5 we summarize and draw conclusions on our findings.

## 2. Methods

We select GRBs from the 2FLGC [8]. From the extensive list of LAT-detected bursts, 86 showed temporally extended emission with a duration ranging from 31 s (for GRB 141102A) to 34,366 s (for GRB 130427A) [48,49,62]. In the catalog, the temporal emission is fitted with a PL function:

$$F(t) = F_0 \left( \frac{t}{T_0} \right)^{-\alpha},$$ (1)

where $F_0$ is the normalization flux, $T_0$ is the trigger time of the GRB, $t$ is the time of observation relative to $T_0$, and $\alpha$ is the temporal decay index, our parameter of interest. GRBs that have four or more measurements of fluxes that are not simply upper limits are additionally fitted in the catalog with a BPL:

$$F(t) = \begin{cases} F_b(\frac{t}{T_b})^{-\alpha_1} & t < T_b \\ F_b(\frac{t}{T_b})^{-\alpha_2} & t \geq T_b, \end{cases}$$ (2)

where $T_b$ is the break time, $F_b$ is the flux at $T_b$, and $\alpha_1$ and $\alpha_2$ are the temporal indices for times before and after $T_b$, respectively.[2] This analysis differs from Tak et al. [50], because they considered the CR with a simple power law, whereas we additionally consider the BPL fitting. For the GRBs for which the BPL fitting is available, we use the BPL fitting from the catalog over the PL model, because the BPL model has two temporal indices, and therefore we can differentiate between the plateau phase for relations with energy injection and the afterglow decay phase for relations without energy injection.

Although in the catalog the flux is determined by integrating between 0.1 and 100 GeV, which would suggest that a smoothly broken power law is more physically accurate to fit the temporally extended emission than the simple BPL, we choose to use the simple BPL results for consistency with the catalog. We estimate that the simple BPL will provide more reliable results than the smooth BPL based on results from a previous study of optical LCs, Dainotti et al. [46], where 179 GRBs were fit with both a simple and smooth BPL. There were 99 GRBs that were successfully fit with the simple BPL, whereas only 45 GRBs were successfully fit with the smooth BPL, corresponding to less than half of the data.

Our sample consists of 86 total GRBs—65 fitted with a PL and 21 fitted with a BPL. Of these 21, 20 GRBs were also present in the sample of Tak et al. [50] and have been reclassified as BPL in our sample. We also consider the BPL, when present, to be consistent with the fitting performed within the 2FLGC, because the BPL can be naturally described by the standard fireball model. When analyzing the CRs, we use the temporal and spectral indices according to the relationship $F_\nu \propto t^{-\alpha} \nu^{-\beta}$. We use the temporal index $\alpha$ for the PL and either $\alpha_1$ or $\alpha_2$ from the catalog for the BPL, depending on the set of CRs being tested. The spectral index $\beta$ is calculated from the photon index $\Gamma$ in the catalog, as $\beta = \Gamma - 1$ (here we define the electron distribution index as $p$). The value of $\Gamma$ is determined in Tak et al. [50] through likelihood analysis during the LAT time window for each GRB, and we assume the determined value to be constant for the entire LC. This assumption is motivated by the fact that at the late time, there is no spectral evolution as indicated in the 2FLGC. More specifically, in the right upper and lower panel of Figure 16 in the 2FLGC Catalog [8], the $\Gamma$ values are shown as a function of the $T_{90,GBM}$ and from the extended emission fluence from the extended time window. The $T_{90,GBM}$ is the $T_{90}$ GRB duration measured by GBM in the 50–300 keV energy range, and the "EXT" interval is defined as the time interval

including LAT emission (if any) after the $T_{GBM,95}$. In both panels of Figure 16 (in Ajello et al. [8]), the green dashed lines denote the mean values, and the dotted lines are the 10% and 90% percentile of each distribution. Thus, we can see that for the majority of bursts, there is no spectral evolution. The values of $\alpha$ and $\beta$ are gathered in Table 1 for the GRBs fitted with a PL and in Table 2 for the GRBs fitted with a BPL. All errors in this work are quoted at the 1 $\sigma$ level.

**Table 1.** Sample of 65 GRBs used here, fitted with a simple power law. Temporal and spectral indices are taken from Ajello et al. [8].

| GRB | $\alpha \pm \delta_\alpha$ | $\beta \pm \delta_\beta$ |
| --- | --- | --- |
| GRB080825C | $1.45 \pm 0.03$ | $1.84 \pm 0.39$ |
| GRB081009A | $0.88 \pm 0.24$ | $0.77 \pm 0.23$ |
| GRB090626A | $0.94 \pm 0.22$ | $1.07 \pm 0.21$ |
| GRB091031A | $1.26 \pm 0.21$ | $1.09 \pm 0.22$ |
| GRB091120A | $0.54 \pm 0.09$ | $1.80 \pm 0.43$ |
| GRB100116A | $2.70 \pm 0.19$ | $0.79 \pm 0.18$ |
| GRB100423B | $0.20 \pm 0.11$ | $0.80 \pm 0.56$ |
| GRB100511A | $0.58 \pm 0.07$ | $0.79 \pm 0.17$ |
| GRB100728A | $0.76 \pm 0.50$ | $0.64 \pm 0.24$ |
| GRB101014A | $-0.16 \pm 0.35$ | $0.31 \pm 0.35$ |
| GRB110120A | $0.55 \pm 0.42$ | $1.03 \pm 0.25$ |
| GRB110428A | $0.98 \pm 0.11$ | $0.93 \pm 0.24$ |
| GRB110518A | $0.72 \pm 0.49$ | $0.80 \pm 0.50$ |
| GRB110625A | $0.57 \pm 0.26$ | $1.74 \pm 0.29$ |
| GRB110721A | $0.95 \pm 0.15$ | $1.43 \pm 0.21$ |
| GRB111210B | $0.55 \pm 0.18$ | $1.67 \pm 0.50$ |
| GRB120226A | $1.21 \pm 0.07$ | $1.94 \pm 0.53$ |
| GRB120316A | $0.87 \pm 0.16$ | $1.16 \pm 0.30$ |
| GRB120526A | $0.69 \pm 0.13$ | $0.84 \pm 0.16$ |
| GRB120624B | $1.19 \pm 0.25$ | $1.53 \pm 0.13$ |
| GRB120709A | $0.67 \pm 0.13$ | $1.30 \pm 0.22$ |
| GRB120711A | $1.63 \pm 0.24$ | $1.06 \pm 0.17$ |
| GRB120911B | $1.31 \pm 0.20$ | $1.47 \pm 0.16$ |
| GRB130325A | $0.13 \pm 0.32$ | $0.51 \pm 0.31$ |
| GRB130502B | $1.44 \pm 0.06$ | $1.03 \pm 0.12$ |
| GRB130518A | $1.09 \pm 0.21$ | $1.85 \pm 0.28$ |
| GRB130606B | $0.67 \pm 0.20$ | $0.74 \pm 0.20$ |
| GRB130821A | $0.99 \pm 0.14$ | $1.37 \pm 0.16$ |
| GRB130828A | $0.96 \pm 0.44$ | $1.24 \pm 0.19$ |
| GRB131014A | $0.82 \pm 0.17$ | $0.90 \pm 0.18$ |
| GRB131029A | $1.11 \pm 0.17$ | $1.40 \pm 0.21$ |
| GRB131209A | $0.83 \pm 0.16$ | $2.27 \pm 0.68$ |
| GRB131231A | $1.03 \pm 0.21$ | $0.67 \pm 0.12$ |
| GRB140102A | $1.22 \pm 0.38$ | $1.15 \pm 0.30$ |
| GRB140104B | $0.30 \pm 0.61$ | $0.98 \pm 0.24$ |
| GRB140110A | $0.97 \pm 0.02$ | $1.64 \pm 0.26$ |
| GRB140206B | $0.30 \pm 0.28$ | $1.09 \pm 0.12$ |
| GRB140402A | $0.87 \pm 0.06$ | $0.76 \pm 0.27$ |
| GRB140523A | $0.96 \pm 0.14$ | $0.99 \pm 0.15$ |
| GRB140810A | $0.82 \pm 0.19$ | $0.53 \pm 0.21$ |
| GRB141028A | $0.97 \pm 0.03$ | $1.44 \pm 0.23$ |
| GRB141102A | $1.03 \pm 0.18$ | $0.91 \pm 0.39$ |
| GRB141207A | $1.88 \pm 0.03$ | $0.80 \pm 0.13$ |
| GRB141222A | $1.33 \pm 0.40$ | $1.10 \pm 0.32$ |
| GRB150314A | $0.95 \pm 0.13$ | $1.50 \pm 0.40$ |
| GRB150523A | $1.03 \pm 0.25$ | $0.92 \pm 0.13$ |
| GRB150702A | $0.44 \pm 0.27$ | $1.23 \pm 0.44$ |

**Table 1.** *Cont.*

| GRB | $\alpha \pm \delta_\alpha$ | $\beta \pm \delta_\beta$ |
|---|---|---|
| GRB150902A | $1.05 \pm 0.18$ | $1.32 \pm 0.18$ |
| GRB160325A | $0.74 \pm 0.10$ | $1.43 \pm 0.24$ |
| GRB160521B | $1.35 \pm 0.20$ | $0.38 \pm 0.26$ |
| GRB160623A | $1.25 \pm 0.09$ | $0.98 \pm 0.11$ |
| GRB160625B | $2.24 \pm 0.28$ | $1.35 \pm 0.07$ |
| GRB160821A | $1.15 \pm 0.10$ | $1.64 \pm 0.20$ |
| GRB160905A | $1.15 \pm 0.28$ | $0.78 \pm 0.16$ |
| GRB161109A | $1.31 \pm 0.48$ | $1.17 \pm 0.33$ |
| GRB170405A | $1.27 \pm 0.01$ | $1.79 \pm 0.35$ |
| GRB170409A | $1.30 \pm 0.11$ | $1.06 \pm 0.28$ |
| GRB170808B | $1.01 \pm 0.22$ | $1.24 \pm 0.27$ |
| GRB170906A | $0.83 \pm 0.12$ | $1.05 \pm 0.14$ |
| GRB171102A | $1.02 \pm 0.07$ | $1.61 \pm 0.58$ |
| GRB171124A | $0.89 \pm 0.09$ | $1.23 \pm 0.24$ |
| GRB171210A | $0.65 \pm 0.28$ | $1.38 \pm 0.29$ |
| GRB180210A | $1.03 \pm 0.19$ | $0.75 \pm 0.14$ |
| GRB180526A | $1.33 \pm 0.70$ | $0.85 \pm 0.28$ |
| GRB180703A | $0.8 \pm 0.15$ | $1.48 \pm 0.32$ |

**Table 2.** Sample of 21 GRBs used in this analysis, fit with a broken power law. Temporal and spectral indices are taken from Ajello et al. [8].

| GRB | $\alpha_1 \pm \delta_{\alpha_1}$ | $\alpha_2 \pm \delta_{\alpha_2}$ | $\beta \pm \delta_\beta$ |
|---|---|---|---|
| GRB080916C | $0.36 \pm 0.29$ | $1.60 \pm 0.21$ | $1.20 \pm 0.06$ |
| GRB090323 | $1.12 \pm 0.17$ | $0.58 \pm 2.44$ | $1.29 \pm 0.15$ |
| GRB090328 | $1.06 \pm 0.14$ | $0.73 \pm 0.43$ | $1.20 \pm 0.13$ |
| GRB090510 | $1.34 \pm 0.18$ | $2.31 \pm 0.17$ | $1.05 \pm 0.06$ |
| GRB090902B | $1.24 \pm 0.23$ | $1.86 \pm 0.16$ | $0.94 \pm 0.04$ |
| GRB090926A | $1.10 \pm 0.17$ | $1.76 \pm 0.16$ | $1.14 \pm 0.05$ |
| GRB091003 | $1.66 \pm 1.08$ | $0.66 \pm 0.39$ | $0.81 \pm 0.16$ |
| GRB100414A | $0.32 \pm 0.55$ | $1.80 \pm 0.21$ | $0.80 \pm 0.12$ |
| GRB110731A | $0.39 \pm 1.07$ | $1.77 \pm 0.11$ | $1.29 \pm 0.16$ |
| GRB130327B | $1.65 \pm 0.28$ | $1.28 \pm 0.77$ | $0.77 \pm 0.11$ |
| GRB130427A | $1.42 \pm 0.10$ | $0.78 \pm 0.16$ | $0.99 \pm 0.04$ |
| GRB130504C | $0.96 \pm 0.49$ | $0.66 \pm 0.11$ | $0.91 \pm 0.20$ |
| GRB131108A | $1.24 \pm 0.62$ | $1.90 \pm 0.17$ | $1.65 \pm 0.11$ |
| GRB150627A | $0.41 \pm 0.07$ | $2.97 \pm 0.24$ | $0.67 \pm 0.12$ |
| GRB160509A | $1.34 \pm 0.90$ | $0.87 \pm 0.26$ | $1.38 \pm 0.12$ |
| GRB160816A | $0.94 \pm 0.61$ | $1.48 \pm 0.33$ | $1.25 \pm 0.17$ |
| GRB170115B | $0.88 \pm 0.67$ | $2.42 \pm 1.78$ | $1.72 \pm 0.41$ |
| GRB170214A | $1.63 \pm 0.46$ | $2.14 \pm 1.82$ | $1.45 \pm 0.09$ |
| GRB171010A | $0.97 \pm 0.29$ | $2.23 \pm 0.73$ | $1.04 \pm 0.13$ |
| GRB171120A | $1.10 \pm 0.59$ | $0.20 \pm 0.57$ | $1.17 \pm 0.19$ |
| GRB180720B | $3.2 \pm 0.56$ | $1.45 \pm 0.18$ | $1.23 \pm 0.10$ |

## 3. Testing the Closure Relations

We test CRs between the indices that correspond to distinct astrophysical environments based on the density profile of the circumburst medium, electron distribution index $p$, and electron cooling regime. We also consider two sets of CRs—one set for a model with energy injection, and the other without. For the GRBs fitted with a PL, we use the $\alpha$ parameter from the 2FLGC for both sets of relations, with and without energy injection. For the GRBs fitted with a BPL, we use the $\alpha_2$ parameter when testing the relations without energy injection. Indeed, these relations test the decay phase of the afterglow that follows the plateau emission but precedes the jet break, the slope of which corresponds to the $\alpha_2$ parameter. When testing relations with energy injection, we use the $\alpha_1$ parameter, which defines the slope of the plateau, as the injection relations specifically test the plateau phase

of the afterglow. Theoretically, the breaks in the long-lasting emission can be interpreted as the passage of the synchrotron cooling break through the Fermi-LAT band [63].

We study several density profiles for both sets of relations, including a constant-density ISM ($n \propto r^0$) and a stratified stellar wind medium ($n \propto r^{-k}$ where $k = 1, 1.5, 2$, and 2.5). We choose these to include the possibility of a stratified medium, in addition to the standard ISM and wind media, and so within the stratified medium, the range of possible $k$ values from $k = 0$ to $k = 2.9$ is evenly sampled. The existence of the stratified medium has been already extensively discussed by Kumar and Piran [21], De Colle et al. [29], Yi et al. [30,31], Crowther [64], De Colle et al. [65], Hotokezaka et al. [66]. We here point out that the same $k$-values are tested for all regimes (fast and slow cooling) and both injection and no-energy injection scenarios. Our inference of the scenarios comes from the testing of different values of $k$. In principle, we could have tested up to $k = 2.9$, but we wish to avoid regions that are close to $k = 3$, as this value is not a real physical value for the CRs, as shown in Yi et al. [30]. Therefore, we choose to test up to 2.5 to be more consistent with physical observations. The set of CRs without energy injection for $k = 0$ and 2 is taken from Tak et al. [50], whereas the set of CRs that consider energy injection for $k = 0$ and 2 is taken from Racusin et al. [54] and Gao et al. [67] . The relations used for other density profiles were derived in this work.

In general, the synchrotron spectral breaks and the maximum synchrotron flux in a stratified medium are $\nu_m \propto \gamma^{4-k} t^{-\frac{k}{2}}$, $\nu_c \propto \gamma^{\frac{3k-4}{2}} t^{-\frac{k}{2}}$, and $F_\nu \propto \gamma^{8-3k} t^{-\frac{3(k-2)}{2}}$, respectively.

Continuous energy injection by the central engine during the GRB afterglow plateau can produce refreshed shocks. The evolution of the bulk Lorentz factor with energy injection is $\gamma \propto t^{-\frac{2-k+q}{8-2k}}$, with $q$ the energy injection index, and, therefore, the synchrotron spectral breaks and the maximum synchrotron flux evolve as $\nu_m \propto t^{-\frac{2+q}{2}}$, $\nu_c \propto t^{-\frac{(2-q)(3k-4)}{2(4-k)}}$, and $F_\nu \propto t^{-\frac{4(k-2)+q(8-3k)}{2(4-k)}}$, respectively. In this case, the closure relations are [23,37,68–71]

$$
F_\nu \propto \begin{cases}
t^{-\frac{[(9k-16)+q(14-6k)]}{3(4-k)}} \left( t^{-\frac{[(7k-16)+q(10-4k)]}{3(4-k)}} \right) \nu^{\frac{1}{3}}, & \nu < \nu_c(\nu_m) \\
t^{-\frac{3q-2}{4}} \nu^{-\frac{1}{2}}, & \nu_c < \nu < \nu_m \\
t^{-\frac{[(2-q)(5k-12)+p(2+q)(4-k)]}{4(4-k)}} \nu^{-\frac{(p-1)}{2}}, & \nu_m < \nu < \nu_c \\
t^{-\frac{[p(2+q)-2(2-q)]}{4}} \nu^{-\frac{p}{2}}, & \max\{\nu_m, \nu_c\} < \nu .
\end{cases}
$$

The luminosity injected from the central engine into the blastwave can be described by $\dot{E}_{k,inj}(t) = L_0 \left( \frac{t}{t_c} \right)^{-q}$, where $q$ controls the rate of energy injection, e.g., [35,37], $L_0$ is the initial luminosity, and $t_c$ is the characteristic timescale. The isotropic-equivalent kinetic energy can be estimated as $E_k = \int \dot{E}_{inj} dt \propto L_0 t^{-q+1}$. A value of $q = 0$ corresponds to continuous energy injection, as in the scenario of the spin-down from a millisecond magnetar [72,73]. These relations are given in the fifth and sixth columns of the bottom half of Table 3. The CRs of the standard synchrotron FS model are recovered for $q = 1$. These relations are given in the fifth and sixth columns of the top half of Table 3.

We study both fast-cooling (characterized by $\nu_c < \nu_m$) and slow-cooling (characterized by $\nu_m < \nu_c$) regimes [23]. It is also possible for the observational frequency to be $\nu > \max\{\nu_c, \nu_m\}$, in which case the CRs do not depend on whether the GRB is in the fast- or slow-cooling regime. Each CR is constrained to a given range of $p$, either $1 < p < 2$ or $p > 2$, which also constrains $\beta$ as a function of $p$ ($\beta = \beta(p)$). Therefore, we consider two regimes, for $1 < p < 2$ and $p > 2$, for the relations without energy injection. For CRs with injection, we only consider $p > 2$, adopting the same treatment as Racusin et al. [54]. We list the explicit relations between the temporal index $\alpha$ and the spectral index $\beta(p)$ in Table 3—in some cases, $\beta(p)$ has a constant value, and the CR is represented as a single point in the plot of $\alpha$ and $\beta$.

We then check if the $\alpha$ and $\beta$ parameters for each GRB, within their error bars, coincide with a given CR. If so, the GRB is considered to obey that CR with a given environment,

cooling regime, and frequency. We assume that the $\alpha$ and $\beta$ parameters are dependent on each other, and therefore expect the errors to be correlated—thus the shape of the error bars are ellipses rather than rectangles. We plot the indices for each GRB with error bars, along with lines representing the CRs for each environment, cooling regime, and frequency range.

We show plots of the complete set of CRs with at least one GRB fulfilling the CR in Figures 1 and 2. The full sample of GRBs is shown in green, whereas GRBs that have indices consistent with the given CR(s) within errors are shown in purple. For the relations both with and without energy injection, the CR is shown as a red line for $p > 2$, as well as a blue line ($1 < p < 2$) for the CRs without energy injection (Figure 1). Constant values of $\beta$ are shown as points. A summary of each CR can be found in Table 3, with the number and proportion of GRBs that satisfy each CR.

**Table 3.** Summary of results of the CRs obtained with observed data without energy injection ($q = 1$; top) and with energy injection ($q = 0$; bottom), showing number and occurrence rate of GRBs satisfying each relation, out of a total of 86 GRBs. As each occurrence rate is computed independently from the total sample of 86 GRBs, the rates are not required to sum to 100%.

| No Energy Injection ($q = 1$) | | | | | | | | |
|---|---|---|---|---|---|---|---|---|
| $n(r)$ | Cooling | $\nu$ Range | $\beta(p)$ | CR:$1 < p < 2$ | CR:$p > 2$ | GRBs | Occurrence Rate | Figure |
| $r^0$ | Slow | $\nu_m < \nu < \nu_c$ | $\frac{p-1}{2}$ | $\frac{6\beta+9}{16}$ | $\frac{3\beta}{2}$ | 22 | 25.6% | (2a) |
| $r^{-1}$ | Slow | $\nu_m < \nu < \nu_c$ | $\frac{p-1}{2}$ | $\frac{4\beta+9}{12}$ | $\frac{9\beta+1}{6}$ | 13 | 15.1% | (2d) |
| $r^{-1.5}$ | Slow | $\nu_m < \nu < \nu_c$ | $\frac{p-1}{2}$ | $\frac{3\beta+9}{10}$ | $\frac{15\beta+3}{10}$ | 10 | 11.6% | (2g) |
| $r^{-2}$ | Slow | $\nu_m < \nu < \nu_c$ | $\frac{p-1}{2}$ | $\frac{2\beta+9}{8}$ | $\frac{3\beta+1}{2}$ | 6 | 7.0% | (2j) |
| $r^{-2.5}$ | Slow | $\nu_m < \nu < \nu_c$ | $\frac{p-1}{2}$ | $\frac{\beta+9}{6}$ | $\frac{9\beta+5}{6}$ | 4 | 4.7% | (2m) |
| $r^0$ | Fast | $\nu_c < \nu < \nu_m$ | $\frac{1}{2}$ | $\frac{\beta}{2}$ | $\frac{\beta}{2}$ | 2 | 2.3% | (2b) |
| $r^{-1}$ | Fast | $\nu_c < \nu < \nu_m$ | $\frac{1}{2}$ | $\frac{\beta}{2}$ | $\frac{\beta}{2}$ | 2 | 2.3% | (2e) |
| $r^{-1.5}$ | Fast | $\nu_c < \nu < \nu_m$ | $\frac{1}{2}$ | $\frac{\beta}{2}$ | $\frac{\beta}{2}$ | 2 | 2.3% | (2h) |
| $r^{-2}$ | Fast | $\nu_c < \nu < \nu_m$ | $\frac{1}{2}$ | $\frac{\beta}{2}$ | $\frac{\beta}{2}$ | 2 | 2.3% | (2k) |
| $r^{-2.5}$ | Fast | $\nu_c < \nu < \nu_m$ | $\frac{1}{2}$ | $\frac{\beta}{2}$ | $\frac{\beta}{2}$ | 2 | 2.3% | (2n) |
| $r^0$ | Slow/Fast | $\nu > \max\{\nu_c, \nu_m\}$ | $\frac{p}{2}$ | $\frac{3\beta+5}{8}$ | $\frac{3\beta-1}{2}$ | 32 | 37.2% | (2c) |
| $r^{-1}$ | Slow/Fast | $\nu > \max\{\nu_c, \nu_m\}$ | $\frac{p}{2}$ | $\frac{\beta+2}{3}$ | $\frac{3\beta-1}{2}$ | 32 | 37.2% | (2f) |
| $r^{-1.5}$ | Slow/Fast | $\nu > \max\{\nu_c, \nu_m\}$ | $\frac{p}{2}$ | $\frac{3\beta+7}{10}$ | $\frac{3\beta-1}{2}$ | 32 | 37.2% | (2i) |
| $r^{-2}$ | Slow/Fast | $\nu > \max\{\nu_c, \nu_m\}$ | $\frac{p}{2}$ | $\frac{\beta+3}{4}$ | $\frac{3\beta-1}{2}$ | 32 | 37.2% | (2l) |
| $r^{-2.5}$ | Slow/Fast | $\nu > \max\{\nu_c, \nu_m\}$ | $\frac{p}{2}$ | $\frac{\beta+5}{6}$ | $\frac{3\beta-1}{2}$ | 31 | 36% | (2o) |
| Energy Injection ($q = 0$) | | | | | | | | |
| $n(r)$ | Cooling | $\nu$ Range | $\beta(p)$ | CR:$p > 2$ | GRBs | Occurrence Rate | Figure | |
| $r^0$ | Slow | $\nu_m < \nu < \nu_c$ | $\frac{p-1}{2}$ | $\beta - 1$ | 16 | 18.6% | (3b) | |
| $r^{-1}$ | Slow | $\nu_m < \nu < \nu_c$ | $\frac{p-1}{2}$ | $\beta - \frac{2}{3}$ | 30 | 34.9% | (3e) | |
| $r^{-1.5}$ | Slow | $\nu_m < \nu < \nu_c$ | $\frac{p-1}{2}$ | $\beta - \frac{2}{5}$ | 40 | 46.5% | (3h) | |
| $r^{-2}$ | Slow | $\nu_m < \nu < \nu_c$ | $\frac{p-1}{2}$ | $\beta$ | 30 | 34.9% | (3k) | |
| $r^{-2.5}$ | Slow | $\nu_m < \nu < \nu_c$ | $\frac{p-1}{2}$ | $\beta + \frac{2}{3}$ | 15 | 17.4% | (3n) | |
| $r^0$ | Fast | $\nu_c < \nu < \nu_m$ | $\frac{1}{2}$ | $-\beta$ | 0 | 0% | (-) | |
| $r^{-1}$ | Fast | $\nu_c < \nu < \nu_m$ | $\frac{1}{2}$ | $-\beta$ | 0 | 0% | (-) | |
| $r^{-1.5}$ | Fast | $\nu_c < \nu < \nu_m$ | $\frac{1}{2}$ | $-\beta$ | 0 | 0% | (-) | |
| $r^{-2}$ | Fast | $\nu_c < \nu < \nu_m$ | $\frac{1}{2}$ | $-\beta$ | 0 | 0% | (-) | |
| $r^{-2.5}$ | Fast | $\nu_c < \nu < \nu_m$ | $\frac{1}{2}$ | $-\beta$ | 0 | 0% | (-) | |
| $r^0$ | Slow/Fast | $\nu > \max\{\nu_c, \nu_m\}$ | $\frac{p}{2}$ | $\beta - 1$ | 16 | 18.6% | (3c) | |
| $r^{-1}$ | Slow/Fast | $\nu > \max\{\nu_c, \nu_m\}$ | $\frac{p}{2}$ | $\beta - 1$ | 16 | 18.6% | (3f) | |
| $r^{-1.5}$ | Slow/Fast | $\nu > \max\{\nu_c, \nu_m\}$ | $\frac{p}{2}$ | $\beta - 1$ | 16 | 18.6% | (3i) | |
| $r^{-2}$ | Slow/Fast | $\nu > \max\{\nu_c, \nu_m\}$ | $\frac{p}{2}$ | $\beta - 1$ | 16 | 18.6% | (3l) | |
| $r^{-2.5}$ | Slow/Fast | $\nu > \max\{\nu_c, \nu_m\}$ | $\frac{p}{2}$ | $\beta - 1$ | 16 | 18.6% | (3o) | |

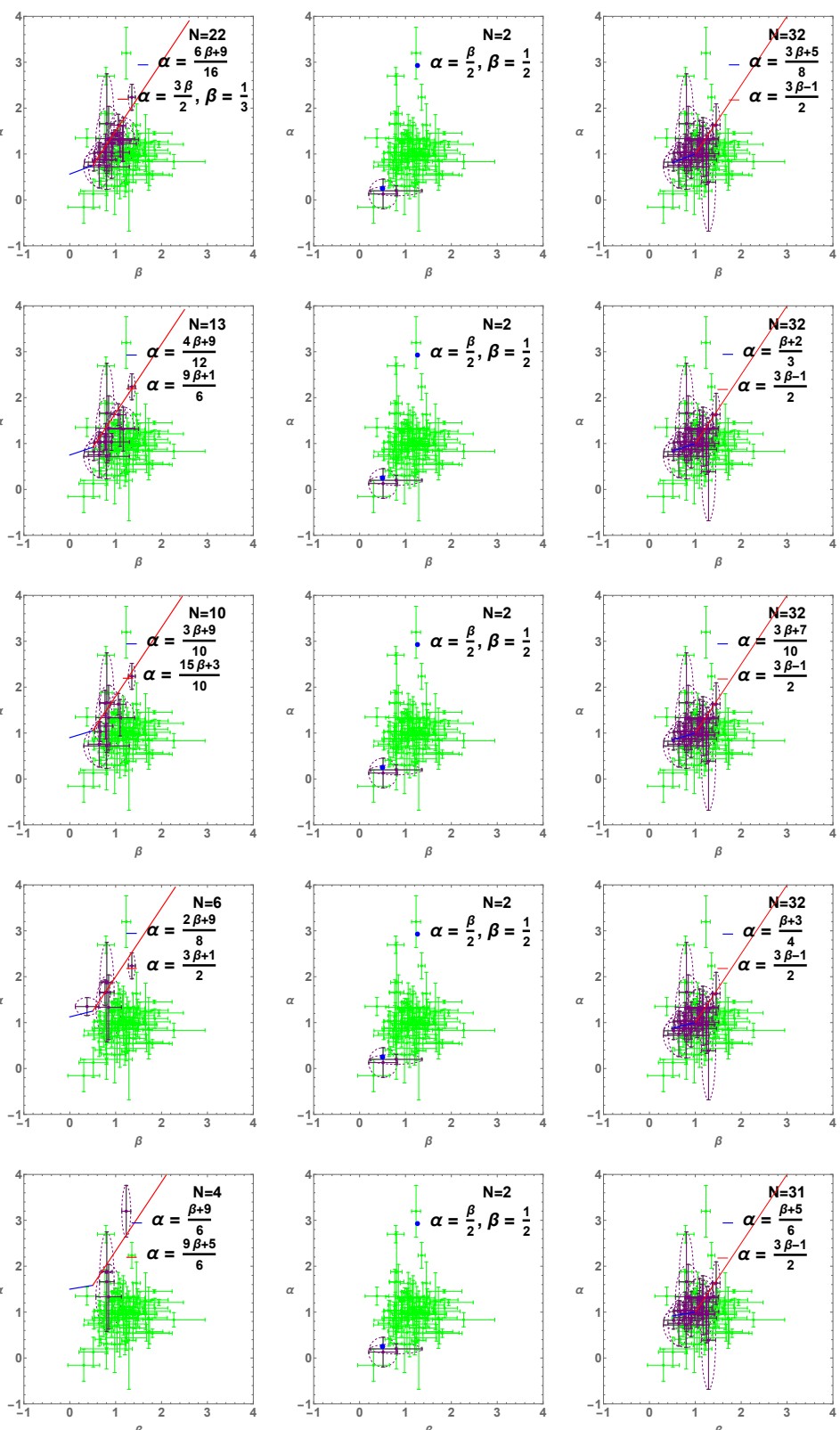

**Figure 1.** CRs from the FS model for $k = 0, 1, 1.5, 2$, and $2.5$ (top row to bottom row, respectively) without energy injection. GRBs that satisfy the relations are shown in purple; other GRBs are shown in green. The first column shows the slow-cooling, $\nu_m < \nu < \nu_c$ regime, the second column shows the fast-cooling, $\nu_c < \nu < \nu_m$ regime, and the third column shows the $\nu > \max\{\nu_c, \nu_m\}$ regime.

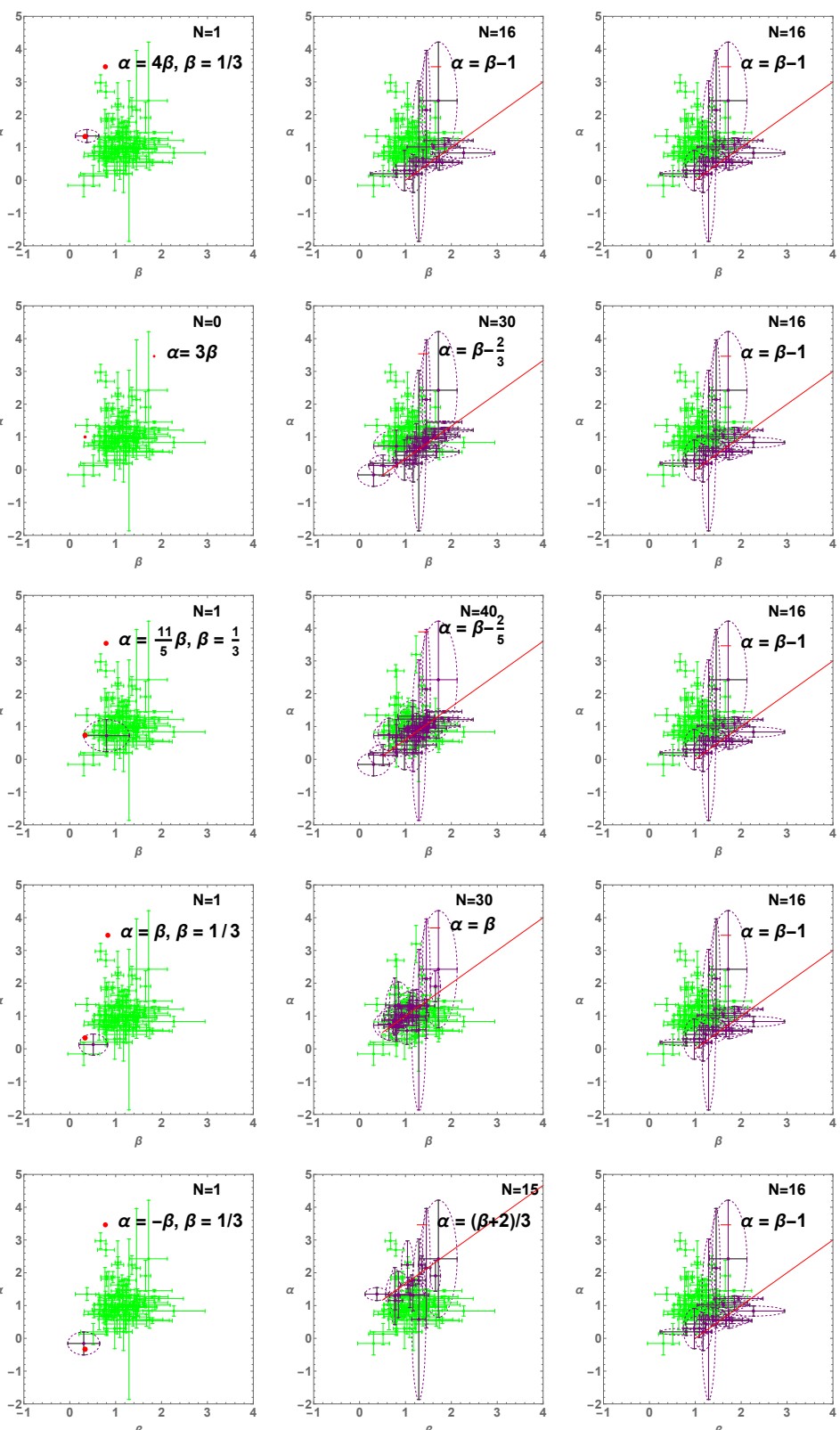

**Figure 2.** CRs from the FS model for $k = 0, 1, 1.5, 2$, and 2.5 (top row to bottom row, respectively) without energy injection and $q = 0$ indicating energy injection. GRBs that satisfy the relations are shown in purple; other GRBs are shown in green. The first column shows the slow-cooling, $\nu_m < \nu < \nu_c$ regime, the second column shows the fast-cooling, $\nu_c < \nu < \nu_m$ regime, and the third column shows the $\nu > \max\{\nu_c, \nu_m\}$ regime.

When considering the relations without energy injection, we find that the $\nu > \max\{\nu_c, \nu_m\}$ regime is the most preferred (see Table 3), with the highest rate of fulfillment in density profiles $k = 0, 1, 1.5, 2$ (37.2%) and with the minimum number of 31 GRBs out of the 86 (36%) GRBs satisfying the case with $k = 2.5$. The second-most preferred environment is the slow-cooling regime in the $\nu_m < \nu < \nu_c$ frequency range, with $k = 0$ showing the highest rate of fulfillment (25.6%) and $k = 2.5$ showing the lowest rate of fulfillment (4.7%). The preference for these two regimes over the fast-cooling relations can be expected, as the $\nu > \max\{\nu_c, \nu_m\}$ and slow-cooling, $\nu_m < \nu < \nu_c$ regimes are represented by a line, naturally intersecting more GRBs, whereas the fast-cooling regime is represented by a point. This makes the fast-cooling, $\nu_c < \nu < \nu_m$ frequency range the least preferred environment, with 2 GRBs (2.33%) for all $k$. There is no significant preference for one value of $k$ over the other.

When considering CRs with energy injection, we find the most preferred environment to be within $\nu_m < \nu < \nu_c$, with the maximum rate of occurrence for $k = 1.5$ with 46.5% and the minimum rate of occurrence being $k = 2.5$ with 17.4% fulfillment. The second-most preferred environment is within the $\nu > \max\{\nu_c, \nu_m\}$ frequency range, with 18.6% fulfillment (16 GRBs) for all values of $k$. The fast-cooling-only regimes are least preferred in all frequency ranges and density profiles, with no fulfillment for all $k$ within $\nu_c < \nu < \nu_m$. Although the highest rate of fulfillment for an individual case is seen in the relations with energy injection (slow-cooling-only, $k = 1.5$ with 46.6% fulfillment), on average, we see that the relations without energy injection are preferred over those with injection.

We also note that the CRs are based on the assumption of a sharply broken power law. In actuality, they would be smoothly broken power laws due to the integration of the flux over the emitting spectrum and the inherent smoothness of the synchrotron spectrum, which may experience further smoothing with the inclusion of energy injection.

As our study is derived from the work in Tak et al. [50], we include a comparison of the best-fulfilled CR for each of the 38 GRBs in common with our sample and that of Tak et al. [50] in Table 4. As Tak et al. [50] studied CRs without energy injection, we compare our results from the cases without energy injection for a more direct comparison. We see that our results match those of Tak et al. [50] exactly for 20/38 GRBs. This means that 18/38 do not agree with Tak et al. [50] results. More specifically, for 5/18 GRBs (GRB 090626A, 091031A, 140810A, 150523A, and 180210A), Tak et al. [50] finds best agreement with the slow-cooling, $\nu_m < \nu < \nu_c$ regime, where we find best agreement with the $\nu > \max\{\nu_c, \nu_m\}$ regime. GRB 180720B satisfies the slow-cooling regime in both studies, but the CR is fulfilled for the ISM ($k = 0$) in Tak and they are fulfilled only with $k = 2.5$ in our study. For 12/18 GRBs in which Tak et al. [50] found agreement with a particular CR, we instead find no fulfillment of any CR in our sample. The differences between our studies arise from the differences in our approaches, as Tak et al. [50] use only PL fitting, whereas in 21 cases we use a BPL. Our approach is deterministic, whereas the Tak et al. [50] approach is probabilistic. Thus, in summary, we can conclude that given these differences overall we have a marginal agreement with Tak et al. [50].

**Table 4.** Comparison between the CR satisfaction of 38 common GRBs between our sample and the analysis done in Tak et al. [50]. The first column is the GRB ID, the second and forth columns are the $\alpha$ for our analysis and Tak et al. [50], respectively. The third and fifth columns show the $\beta$ indices for our analysis and Tak et al. [50]. The sixth and seventh columns show the fulfilled CR in Tak et al. [50] and its corresponding environment, whereas the eighth and ninth columns show the fulfilled CR and its corresponding environment in our study. The last column marks agreement with Tak et al. [50]. Data from Tak et al. [50] are marked with $^T$.

| GRB ID | $\alpha \pm \delta_\alpha^T$ | $\beta \pm \delta_\alpha^T$ | $\alpha \pm \delta_\alpha$ | $\beta \pm \delta_\beta$ | $CR^T$ | ISM/Wind $^T$ | $CR$ | k Values | Match |
|---|---|---|---|---|---|---|---|---|---|
| GRB090626A | $0.94 \pm 0.22$ | $1.00 \pm 0.21$ | $0.94 \pm 0.22$ | $1.07 \pm 0.21$ | $\nu_m < \nu < \nu_c$ | ISM | $\nu > \nu_m, \nu_c$ | $k = 0 - 2.5$ | No |
| GRB091031A | $1.26 \pm 0.21$ | $0.81 \pm 0.23$ | $1.26 \pm 0.21$ | $1.09 \pm 0.22$ | $\nu_m < \nu < \nu_c$ | ISM | $\nu > \nu_m, \nu_c$ | $k = 0 - 2.5$ | No |
| GRB100116A | $2.70 \pm 0.19$ | $0.59 \pm 0.18$ | $2.70 \pm 0.19$ | $0.79 \pm 0.18$ | None | None | None | None | Yes |
| GRB100511A | $0.58 \pm 0.07$ | $0.75 \pm 0.17$ | $0.58 \pm 0.07$ | $0.79 \pm 0.17$ | None | None | None | None | Yes |
| GRB110625A | $0.57 \pm 0.26$ | $1.67 \pm 0.26$ | $0.57 \pm 0.26$ | $1.74 \pm 0.29$ | None | None | None | None | Yes |
| GRB120526A | $0.69 \pm 0.13$ | $0.87 \pm 0.16$ | $0.69 \pm 0.13$ | $0.84 \pm 0.16$ | None | None | None | None | Yes |
| GRB120624B | $1.19 \pm 0.25$ | $1.53 \pm 0.13$ | $1.19 \pm 0.25$ | $1.53 \pm 0.13$ | None | None | None | None | Yes |
| GRB120709A | $0.67 \pm 0.13$ | $1.42 \pm 0.32$ | $0.67 \pm 0.13$ | $1.30 \pm 0.22$ | None | None | None | None | Yes |
| GRB120711A | $1.63 \pm 0.24$ | $1.08 \pm 0.17$ | $1.63 \pm 0.24$ | $1.06 \pm 0.17$ | $\nu_m < \nu < \nu_c$ | ISM | $\nu_m < \nu < \nu_c$ | $k = 0 - 1.5$ | Yes |
| GRB120911B | $1.31 \pm 0.20$ | $1.33 \pm 0.30$ | $1.31 \pm 0.20$ | $1.47 \pm 0.16$ | $\nu > \nu_m, \nu_c$ | ISM/Wind | None | None | No |
| GRB130325A | $0.13 \pm 0.32$ | $0.69 \pm 0.28$ | $0.13 \pm 0.32$ | $0.51 \pm 0.31$ | $\nu_c < \nu < \nu_m$ | ISM/Wind | $\nu_c < \nu < \nu_m$ | $k = 0 - 2.5$ | Yes |
| GRB130502B | $1.44 \pm 0.06$ | $0.99 \pm 0.14$ | $1.44 \pm 0.06$ | $1.03 \pm 0.12$ | $\nu_m < \nu < \nu_c$ | ISM | $\nu_m < \nu < \nu_c$ | $k = 0$ | Yes |
| GRB130518A | $1.09 \pm 0.21$ | $1.73 \pm 0.30$ | $1.09 \pm 0.21$ | $1.85 \pm 0.28$ | None | None | None | None | Yes |
| GRB130606B | $0.67 \pm 0.20$ | $0.74 \pm 0.20$ | $0.67 \pm 0.20$ | $0.74 \pm 0.20$ | $\nu_m < \nu < \nu_c$ | ISM | None | None | No |
| GRB130821A | $0.99 \pm 0.14$ | $1.35 \pm 0.25$ | $0.99 \pm 0.14$ | $1.37 \pm 0.16$ | $\nu > \nu_m, \nu_c$ | ISM/Wind | None | None | No |
| GRB131014A | $0.82 \pm 0.17$ | $0.97 \pm 0.25$ | $0.82 \pm 0.17$ | $0.90 \pm 0.18$ | $\nu > \nu_m, \nu_c$ | ISM/Wind | $\nu > \nu_m, \nu_c$ | $k = 0 - 2.5$ | Yes |
| GRB131029A | $1.11 \pm 0.17$ | $1.27 \pm 0.33$ | $1.11 \pm 0.17$ | $1.40 \pm 0.21$ | $\nu > \nu_m, \nu_c$ | ISM/Wind | None | None | No |
| GRB131231A | $1.03 \pm 0.21$ | $0.63 \pm 0.12$ | $1.03 \pm 0.21$ | $0.67 \pm 0.12$ | $\nu_m < \nu < \nu_c$ | ISM | $\nu_m < \nu < \nu_c; \nu > \nu_m, \nu_c$ | $k = 0 - 2.5$ | Yes |
| GRB140206B | $0.30 \pm 0.28$ | $0.95 \pm 0.13$ | $0.30 \pm 0.28$ | $1.09 \pm 0.12$ | None | None | None | None | Yes |
| GRB140523A | $0.96 \pm 0.14$ | $1.03 \pm 0.19$ | $0.96 \pm 0.14$ | $0.99 \pm 0.15$ | $\nu > \nu_m, \nu_c$ | ISM/Wind | $\nu > \nu_m, \nu_c$ | $k = 0 - 2.5$ | Yes |
| GRB140810A | $0.82 \pm 0.19$ | $0.55 \pm 0.21$ | $0.82 \pm 0.19$ | $0.53 \pm 0.21$ | $\nu_m < \nu < \nu_c$ | ISM | $\nu > \nu_m, \nu_c$ | $k = 0 - 2.5$ | No |
| GRB141028A | $0.97 \pm 0.03$ | $1.01 \pm 0.25$ | $0.97 \pm 0.03$ | $1.44 \pm 0.23$ | $\nu > \nu_m, \nu_c$ | ISM/Wind | None | None | No |
| GRB141207A | $1.88 \pm 0.03$ | $0.80 \pm 0.30$ | $1.88 \pm 0.03$ | $0.80 \pm 0.13$ | $\nu_m < \nu < \nu_c$ | Wind | $\nu_m < \nu < \nu_c$ | $k = 2, 2.5$ | No |
| GRB141222A | $1.33 \pm 0.40$ | $1.10 \pm 0.33$ | $1.33 \pm 0.40$ | $1, 10 \pm 0.32$ | $\nu > \nu_m, \nu_c$ | ISM/Wind | $\nu > \nu_m, \nu_c; \nu_m < \nu < \nu_c$ | $k = 0 - 2.5$ | Yes |
| GRB150523A | $1.03 \pm 0.25$ | $0.78 \pm 0.16$ | $1.03 \pm 0.25$ | $0.92 \pm 0.13$ | $\nu_m < \nu < \nu_c$ | ISM | $\nu > \nu_m, \nu_c$ | $k = 0 - 2.5$ | No |
| GRB150902A | $1.05 \pm 0.18$ | $1.06 \pm 0.20$ | $1.05 \pm 0.18$ | $1.32 \pm 0.18$ | $\nu > \nu_m, \nu_c$ | ISM/Wind | None | None | No |
| GRB160325A | $0.74 \pm 0.10$ | $1.40 \pm 0.24$ | $0.74 \pm 0.10$ | $1.43 \pm 0.24$ | None | None | None | None | Yes |
| GRB160521B | $1.35 \pm 0.20$ | $0.41 \pm 0.26$ | $1.35 \pm 0.20$ | $0.38 \pm 0.26$ | $\nu_m < \nu < \nu_c$ | Wind | $\nu_m < \nu < \nu_c$ | $k = 2$ | Yes |
| GRB160623A | $1.25 \pm 0.09$ | $0.98 \pm 0.11$ | $1.25 \pm 0.09$ | $0.98 \pm 0.11$ | $\nu_m < \nu < \nu_c$ | ISM | None | None | No |

**Table 4.** *Cont.*

| GRB ID | $\alpha \pm \delta_\alpha^T$ | $\beta \pm \delta_\alpha^T$ | $\alpha \pm \delta_\alpha$ | $\beta \pm \delta_\beta$ | $CR^T$ | ISM/Wind $^T$ | $CR$ | k Values | Match |
|---|---|---|---|---|---|---|---|---|---|
| GRB160625B | $2.24 \pm 0.28$ | $0.75 \pm 0.28$ | $2.24 \pm 0.28$ | $1.35 \pm 0.07$ | $\nu_m < \nu < \nu_c$ | Wind | $\nu_m < \nu < \nu_c$ | $k = 0 - 2$ | Yes |
| GRB160821A | $1.15 \pm 0.10$ | $0.75 \pm 0.17$ | $1.15 \pm 0.10$ | $1.64 \pm 0.20$ | $\nu_m < \nu < \nu_c$ | ISM | None | None | No |
| GRB160905A | $1.15 \pm 0.28$ | $0.84 \pm 0.19$ | $1.15 \pm 0.28$ | $0.78 \pm 0.16$ | $\nu_m < \nu < \nu_c$ | ISM | $\nu_m < \nu < \nu_c; \nu > \nu_m, \nu_c$ | $k = 0 - 2.5$ | Yes |
| GRB170405A | $1.27 \pm 0.01$ | $1.51 \pm 0.33$ | $1.27 \pm 0.01$ | $1.79 \pm 0.35$ | $\nu > \nu_m, \nu_c$ | ISM/Wind | None | None | No |
| GRB170808B | $1.01 \pm 0.22$ | $1.15 \pm 0.26$ | $1.01 \pm 0.22$ | $1.24 \pm 0.27$ | $\nu > \nu_m, \nu_c$ | ISM/Wind | $\nu > \nu_m, \nu_c$ | $k = 0 - 2.5$ | Yes |
| GRB170906A | $0.83 \pm 0.12$ | $1.06 \pm 0.14$ | $0.83 \pm 0.12$ | $1.05 \pm 0.14$ | $\nu > \nu_m, \nu_c$ | ISM/Wind | None | None | No |
| GRB180210A | $1.03 \pm 0.19$ | $0.74 \pm 0.14$ | $1.03 \pm 0.19$ | $0.75 \pm 0.14$ | $\nu_m < \nu < \nu_c$ | ISM | $\nu > \nu_m, \nu_c$ | $k = 0 - 2.5$ | No |
| GRB180703A | $0.85 \pm 0.15$ | $1.38 \pm 0.33$ | $0.8 \pm 0.15$ | $1.48 \pm 0.32$ | $\nu > \nu_m, \nu_c$ | ISM/Wind | None | None | No |
| GRB180720B | $1.88 \pm 0.15$ | $1.15 \pm 0.10$ | $3.2 \pm 0.56$ | $1.45 \pm 0.18$ | $\nu_m < \nu < \nu_c$ | ISM | $\nu_m < \nu < \nu_c$ | $k = 2.5$ | No |

## 4. Comparing to CRs in Other Bands

To gain a more complete understanding of the fulfillment of the standard fireball model among different wavelengths, we compare the fulfillment of our sample to the fulfillment of CRs in X-ray, optical, and radio bands. We investigate five GRBs in common with the X-ray sample in Srinivasaragavan et al. [55], one GRB in common with the optical sample in Dainotti et al. [74], and one GRB in common with both samples, which have been tested against the same set of CRs in this work for the $k = 0$ and $k = 2$ cases, without energy injection. Therefore, we only consider the comparison with our sample for these relations, for consistency. We also consider a comparison with one GRB in common with the radio sample in Levine et al. [75] in the same regimes, although the CRs are different at these lower energies. The result of this comparison is given in Table 5.

**Table 5.** Table summarizes fulfillment of CRs in multiple wavelengths. Column 1 gives the name of the GRB in the high-energy sample that has also been analyzed in either X-rays, optical, or radio wavelengths. Column 2 gives the wavelengths for which a particular GRB has been analyzed—$\gamma$ for $\gamma$-rays, X for X-rays, O for optical, and R for radio. Columns 3, 4, and 5 mark whether a GRB satisfies the given CR in the ISM ($k = 0$) environment, while columns 6, 7, and 8 mark whether a particular GRB satisfies the given CR in the wind ($k = 2$) environment. Columns are marked with X, O, or R to represent whether the relation has been satisfied in X-ray, optical, or radio wavelengths; '...' is placed in all columns for which the GRB does not satisfy the given relation.

| | | ISM | | | Wind | | |
|---|---|---|---|---|---|---|---|
| **GRB** | **Wavelength** | $\nu_m < \nu < \nu_c$ | $\nu_c < \nu < \nu_m$ | $\nu > \max\{\nu_c, \nu_m\}$ | $\nu_m < \nu < \nu_c$ | $\nu_c < \nu < \nu_m$ | $\nu > \max\{\nu_c, \nu_m\}$ |
| GRB090510 | $\gamma, O, X$ | ... | ... | O | ... | ... | O |
| GRB100728A | $\gamma, X$ | $\gamma$ | ... | $\gamma$ | ... | ... | $\gamma$ |
| GRB110731A | $\gamma, X$ | ... | ... | $\gamma$ | ... | ... | $\gamma$ |
| GRB120711A | $\gamma, O$ | $\gamma$ | ... | ... | ... | ... | ... |
| GRB150314A | $\gamma, X$ | ... | ... | ... | ... | ... | ... |
| GRB160509A | $\gamma, R$ | ... | ... | $\gamma$ | ... | ... | $\gamma$ |
| GRB170405A | $\gamma, X$ | ... | ... | ... | ... | ... | ... |
| GRB180720B | $\gamma, X$ | ... | ... | ... | ... | ... | ... |

We see that none of the GRBs fulfill the fast-cooling relations in either an ISM or a wind environment for any wavelength, indicating consistency among bands. Overall, the results indicate a preference for the $\nu > \max\{\nu_c, \nu_m\}$ regime among wavelengths. Regarding individual GRBs, for the five GRBs analyzed in $\gamma$- and X-rays, GRB 100728A satisfies the slow-cooling, $\nu_m < \nu < \nu_c$ regime for the ISM environment and the $\nu > \max\{\nu_c, \nu_m\}$ regime in both the ISM and wind environments for $\gamma$-rays. GRB 110731A satisfies the $\nu > \max\{\nu_c, \nu_m\}$ regime in $\gamma$-rays for both an ISM and a wind environment, whereas GRB 150314A and GRB180720B satisfy none of the given relations in either band. The GRB coincident with the optical sample, GRB 120711A, satisfies the slow-cooling, $\nu_m < \nu < \nu_c$ regime for an ISM environment in $\gamma$-rays only. GRB 090510 has been tested in X-ray, optical, and $\gamma$-ray bands, and is found to satisfy the $\nu > \max\{\nu_c, \nu_m\}$ regime in optical wavelengths for both an ISM and a wind environment. GRB 160509A, which has been tested in $\gamma$-rays and radio wavelengths, satisfies none of the given relations.

## 5. Summary and Conclusions

We have tested the external FS model using CRs with high-energy GRBs from the 2FLGC. With a better understanding of the physical characteristics of GRBs, which could be highlighted by CR studies, we could improve our ability to classify and standardize them in the future.

A significant portion of our sample of 86 GRBs satisfies the CRs—79 GRBs in our sample satisfy at least one CR—suggesting that the external FS model can explain many of the characteristics found in high-energy GRBs. We note that when a GRB fulfills one CR it means that both the energy mechanism and the environment can be determined. When

a GRB fulfills more than one CR it means that we have equally probable scenarios. We cannot neglect that in many cases we have degeneracy of scenarios. Nevertheless, this is still important, because our results still allow some scenarios to be removed from the picture. When we have multiple wavelengths we can refer to the physical meaning and interpretation.

For a more detailed comparison on a one-to-one GRB basis, we compare our results to prior results in X-ray, optical, and radio bands when GRBs are observed simultaneously in multiple bands. We find that, similar to the high-energy results, the slow-cooling, $\nu_c < \nu < \nu_m$ and $\nu > \max\{\nu_c, \nu_m\}$ regimes are preferred in other bands, whereas the fast-cooling, $\nu_c < \nu < \nu_m$ is not fulfilled by any of the bands.

If we take the average of the fulfillment rates for each case, we find that the CRs without energy injection are generally preferred over those that assume energy injection.

When considering CRs without energy injection, we find that 43 of the GRBs in our sample satisfy at least one CR used in this study. We here note that the number of GRBs in this computation is not repeated, whereas the GRBs indicated in the table consider also repeated GRBs, but not in the same CR. When we consider the subsample of the 21 GRBs fitted with a BPL, we find 9 GRBs (GRB 090926A, GRB 091003A, GRB 110731A, GRB 130504C, GRB 160509A, GRB 160816A, GRB 170214A, GRB 171010A, and GRB 171120A) to have $\alpha$ and $\beta$ parameters consistent with the $\nu > \max\{\nu_c, \nu_m\}$ regime for all values of $k$. GRB 091003A fulfills the slow-cooling, $\nu_m < \nu < \nu_c$ regime for all values of $k$. Three GRBs (090902B, 130427A, and 130504C) fitted with a BPL fulfill this regime for $k = 0$, whereas 130327B fulfills this regime for $k = 1.5, 2$, and GRB 180720B fulfills this regime for $k = 2.5$. The fast-cooling, $\nu_c < \nu < \nu_m$ regime is not satisfied by any GRBs to fit with a BPL for any values of $k$ without energy injection.

For the CRs with energy injection, we find that only 15 GRBs do not satisfy any of the CRs. Four GRBs with a BPL (GRB 090323A, GRB 170115A, GRB 170214A, and GRB 171120A) satisfy the $\nu > \max\{\nu_c, \nu_m\}$ regime for all $k$. In the slow-cooling, $\nu_m < \nu < \nu_c$ regime, GRBs 090323A, 170115B, and 170214A satisfy the relations for all values of $k$. GRB 1701120A fulfills the $k = 0, 1$, and $k = 1.5$ profiles. GRBs 160509A and 090328 fulfill the $k = 1, 1.5$ profiles; GRBs 130327B and 131108A fulfill the $k = 2, 2.5$ profiles; GRB 091003A fulfills the $k = 1.5, 2$ profiles. The $k = 1.5$ profile is additionally fulfilled by GRB 130504C, $k = 2$ profile is additionally fulfilled by GRB 160816A, and the $k = 2.5$ profile is additionally fulfilled by three unique GRBs—GRB 090926A, GRB 110731A, and GRB 171010A. The fast-cooling-only regime is not fulfilled by any GRBs with a BPL.

We find many similarities in our results with those of Tak et al. [50]. The authors use a sample of 59 GRBs from the 2FLGC and find that the most preferred environments are the slow-cooling regime without energy injection within the $\nu_m < \nu < \nu_c$ frequency range and the $\nu > \max\{\nu_c, \nu_m\}$ range. This agrees with our data for relations without energy injection. They find the least agreement within the fast-cooling, $\nu_c < \nu < \nu_m$ regime, which also agrees with our results.

Regarding the CRs without energy injection, we find that, within the slow-cooling, $\nu_m < \nu < \nu_c$ regime, the $k = 0$ profile is favored in our sample. This seems to disagree with Tak et al. [50], as well as previous work in Srinivasaragavan et al. [55] and Dainotti et al. [56], which concluded that a wind environment ($k = 2$) is most favored. For relations with energy injection, we see that the $k = 1.5$ profile within this regime is most preferred. The differences between the two studies may come from the larger sample (86 vs. 59 in Tak et al. [50], 31% larger) and the set of $k$ values beyond the traditionally assumed values $k = 0, 2$. In addition, we classify CR satisfaction using a frequentist approach rather than a Bayesian probability, so our agreement despite the differing approaches strengthens the conclusions of both papers. We find that of the 38 GRBs that are common with the ones mentioned in Tak et al. [50], 20 GRBs agree with Tak et al. [50], and out of the 18/38 which do not agree, 5/18 prefer a different frequency range, 1/18 prefers a different environment within the same frequency range, and 12/18 do not fulfill any relation in our study.

We also notice that the temporal indices ($\alpha$) and spectral index ($\beta$) values are similar to those noted in the data sample of Tak et al. [50]. We here note that, in another high-energy study by Dainotti et al. [56], the slow-cooling regime with energy injection is also preferred. However, the studies performed in Srinivasaragavan et al. [55] without energy injection show that there is a disagreement between CRs at high and low energies, suggesting that the emission mechanism may differ at high energies for those without energy injection. This idea is supported by a study of 82 optical GRBs, in which Dainotti et al. [74] found that when testing a set of CRs, the majority of the sample did not fulfill any CR, indicating disagreement with the standard fireball model.

It is possible that the emission mechanism in lower energies could be described by SSC emission. Regarding the theoretical interpretation, the breaks in the long-lasting emission can be explained with the passage of the synchrotron cooling break through the Fermi-LAT band [76]. We note that the spectral index is independent of the energy injection, so the assumption of the non-spectral evolution is valid in the case of a temporal break resulting from the turning-off energy injection. However, because we interpret a temporal break as the passage of the cooling break, the constant spectral index may not be valid; when a cooling break is passed, the spectral index should change as described in Table 3.

The preference $\nu_{LAT} > \nu_{m,c}$ is due to the fact that, in the case of several interstellar media (e.g., for a particular case of ISM), the spectral breaks evolve as $\nu_m \propto t^{-\frac{3}{2}}$ and $\nu_c \propto t^{-\frac{1}{2}}$. Therefore, for late-time observations (e.g., where $t_d$, the deceleration time, is $\approx 10^3$ seconds, and in this case we assume $T_{break} = t_d$), the LAT frequency is expected to be in the cooling regime $\nu_{LAT} > \max\{\nu_c, \nu_m\}$. In this case, the magnetic microphysical parameter ($\epsilon_B$) would be constrained by the following equation (Equation (11) in [23], for the adiabatic regime):

$$\epsilon_B > 2.43 \times 10^{-8} n^{-\frac{2}{3}} E_{52\mathrm{erg}}^{-\frac{1}{3}} t_d^{-\frac{1}{3}} \left( \frac{h\nu_c}{100\,\mathrm{MeV}} \right)^{-\frac{2}{3}}, \tag{3}$$

where $E$ is the equivalent kinetic energy, and $n$ is the density medium.

Ultimately, these CRs are a rudimentary test of the current external FS model, which depends on various assumed characteristics. Further study of these CRs using more complex models could help establish classifications and standardize GRBs.

**Author Contributions:** Conceptualization: M.D.; Methodology: M.D. and N.F.; Software: M.D.; Validation: M.D. and D.L.; Formal analysis: D.L.; Investigation, D.L. and M.D.; Data curation: M.D. and D.L.; Writing—original draft preparation: D.L. and M.D.; Writing—review and editing: M.D., N.F., D.L., D.W., P.V., and S.S.; Supervision: M.D. and N.F. All authors have read and agreed to the published version of the manuscript.

**Funding:** This research received no external funding.

**Institutional Review Board Statement:** Not applicable.

**Data Availability Statement:** Data used in the current analysis have been taken from the Second Fermi GRB LAT Catalog as note in the data sample.

**Acknowledgments:** We are grateful to Peter Richardson for his initial work on the CRs in high energy and Kevin Zvonarek for his discussion of CRs. We also thank Aleksander Lenart for his work on the initial realization of the CR notebook. We also thank Seth Diegel for his comments and suggestions in writing this paper. This work was supported in part by the U.S. Department of Energy, Office of Science, and Office of Workforce Development for Teachers and Scientists (WDTS) under the Science Undergraduate Laboratory Internships (SULI) program. NF acknowledges financial support from UNAM-DGAPA-PAPIIT through grant IN106521. We also acknowledge the support of the NAOJ Division of Science and the Exploratory Research Fund in making this research possible.

**Conflicts of Interest:** The authors declare no conflict of interest. The founding sponsors had no role in the design of the study; in the collection, analysis, or interpretation of data; in the writing of the manuscript, and in the decision to publish the results.

## Notes

<sup></sup>

1     Where the cooling timescale for shocked electrons is comparable to, or longer than, the dynamic timescale of the GRB jet.

2     For a sample visualization of these fittings, see Figure 9 of Ajello et al. [8].

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
