# Peer review of "The Closure Relations in High-Energy Gamma-ray Bursts Detected by Fermi-LAT"

_galaxies, doi:10.3390/galaxies11010025_

Round 1

Reviewer 1 Report

Dear authors, I would like to suggest you a few changes. The most important one is to differentiate the discussion on the closure relations obtained in other bands with the one obtained here. It's not clear to me wheter the CR are exactly the same in different portions of the spectral energy distribution. 

The second change, is once you have obtained the number of GRBs that fulfill a certain CR, what could be learned? Could the verified CR tell something important for that GRB? Could the CR be tested against other observations? (for example in other wavelenghts for the same GRBs)? 

A general comment on the references, the journal name is often missing. 

Another general comment, the symbol \nu is often missing (e.g. in L11)

Specific comments: 

L29: is 20 or 30 MeV the lower limit for LAT? 

L32: when is the starting date for the second LAT GRB catalog? 

L37: how a GRB before the LAT launch (080319) could be included in the LAT catalog? 

L78: what is the "charateristic" frequency? 

L81: Why the high-energy could be "naturally" explained by SSC? Are the closure relations different in that case? 

L123: Here you discuss 190114C. I would argue well before if your test on the CR is for the afterglow or the prompt emission, and in that case to explain how you defined that. 

L127: Are the values of k physically motivated? How? 

L133: Please be consistent with the GRB nomenclature. 

L139: Here you introduce the "plateau" and "afterglow" phase. I would have moved, as said before, this discussion earlier, arguing somehow the same shape of the light curve in X-rays could be adopted for gamma-rays. 

LL150-161: this statistical discussion seems out of the context. It might be important to use that provided you showed some of the tests you might have performed, otherwise it seems just a claim without evidence. 

L204: why k=2.5 is still ok and k=3 is not? What are the physically motivated explanations for k? 

L210: Introduce here the explanation for q. In the paper it appears two pages later .. 

L215: Why you don't include the case q=1? 

L242: The sentence is going out of the text limits. 

L267: Either "more" or "less" fulfillment. What does it mean? Are your results solid or not? 

L272: How do you explain the misagreement with Tak et al. Are your results marginally consistent? Do you or they use a different expression for CR? It's strange that based on the same results (the 2nd GRB catalog) two groups obtain such a large difference. You say something in lines LL322-324 but I cannot understand your point there. 

L285 and L290: what does "at least one CR" .. does it mean that in this case you can measure the mechanism and the environment? How could a GRB fulfill more than one CR? If it's the case, what is the usefulness of such relations? 

L338: Here you write that SSC explains the emission at "lower" energies .. check with your statement in line 81. 

Reviewer 2 Report

A very interesting paper. I recommend publication, however I have a few comments.

Abstract: line 11: I do not understand this line, maybe there is typo in the \nu . See also e.g., line 218-219 and following pages

line 56: closure relationships-> closure relations

line 57: Closure relations exist everywhere in different fields. Perhaps you mean that within the standard fireball model, they are equations that define ...

line 66: I strongly recommend to consider also the works of Gompertz+18, ApJ866, and the seminal work of Schulze+11, A&A 526, although they concentrate on the afterglow phase.

line 87: FC and SC are regimes, not environments.

line 142: Do you mean the BPL to fit the spectra? If so, please specify and clarify why one should suggest this.

Reviewer 3 Report

The article entitled with "The Closure Relations in High-Energy Gamma-Ray Bursts detected by Fermi-LAT" considered the Closure Relations for 186 GRB afterglows from Fermi-LAT Second GRB Catalog. They fitted the light curve power law indices, and the spectral indices. They used the standard afterglows model with synchrotron radiation, by considering circum burst environment as a pseudo-free parameter k, with given certain choices; and considering with or without energy injection, to fit the closure relations. It is worth to be published if they can address the following comment. 

What is the critereion for them to choose PL or BPL for a certain GRB? It might be explicitly explained, likely before line 162. 

In line 168, the authors said for the BPL, they use alpha1 and alpha2. However, in Table 2, they use alpha and alpha1.

It is not very easy to understand how they get the parameters, such as alpha, beta, etc. I would suggest the author show one or two figures as examples, for the fitting parameters shown in Tables 1 and 2. That may show the readers a better senario what they are doing. 

In Table 2, they gave only one beta. However, in the GRB afterglows models, for broken power laws, the spectral indices are generally different. Why the authors just provide only one beta for each GRB? 

line 214, I am confused about the explanation: "the closure relations due to spin-down from a millisecond magnetar are obtained for q = 0" Instantaneous energy injection means abrupt energy injection, that is the standard fireball model. However, this is not the case for spin-down from a millisecond magnetar. Can the authors clarify that?

line 225, "only consider p > 2", why?

Some minor suggestions:

line 175, when they mention Fig. 16 from a reference, the reference should be shown.

line 210, the references of the closure relations should be shown.

below that line, they also should give the references for the formulas. Especially for the case with energy injection. And the author may also explicitly write down the formula for energy injection, such as Ek \propto t^-q.

I would suggest the authors reduce the abbreviations. It is really a pain to remember those abbreviations. I would suggest to adopt normal words for those not often used one, such as: FC, SC, LC.

The authors may carefully reread their own paper, and correct the obivous typos, such as but not limited to:

line 11, and many other places, the frequency $\nu$ is not correctly displayed.

Eq. (3), \hat{L} is different from the discription in line 156, L.

Reviewer 4 Report

The authors test the theoretical closure relations for a general stratified circumburst medium and astrophysical environments. This is an interesting work concerning the closure relations in high-energy GRBs. I have two comments below:

(1) Since the BPL and PL models have different numbers of free parameters, a comparison of the likelihoods judging which model is a better match to the data must be based on the AIC or BIC statistical criteria. The authors fitted 21 GRBs with a smooth BPL. However, I did not find a text to explain why BPL is preferred over PL for these 21 GRBs. The authors should describe the outcome of their AIC or BIC analysis somewhere.

(2) The authors may want to make the size of the letters and numbers in the figures more consistent and generally larger. The labels in the plots are too small.

Round 2

Reviewer 1 Report

Thanks for implementing my comments. I am satisfied. 

Reviewer 3 Report

The authors have fully addressed all the comments. I would suggest it to be accepted by Galaxies.